# Random Forest Regression and Shapley Additive Explanation for Effective Dose Rate Estimation in High-Energy Neutron Fields Based on Bonner Spectrometer Measurements

**Konstantin Chizhov**
Meshcheryakov Laboratory of Information Technology
Joint Institute for Nuclear Research,
Dubna, Russia.
University "Dubna",
Dubna, Russia.
kchizhov@jinr.ru

**Belyi Artem & Starikovskaya Maria**
University "Dubna",
Dubna, Russia.

## Abstract

The article proposes a method for assessing the neutron energy spectrum and effective dose rate of personnel based on the readings of a Bonner spectrometer (BSS) for high-energy neutron fields. Neutron flux density can be obtained from BSS measurements by solving the system of Fredholm integral equations of the first kind. In our paper the spectra were unfolded using supervised machine learning algorithm "random forest" with optimization of the model hyperparameters. The model was trained and tested on a database of 251 spectra for various power facilities (80% of data was used for training the model, while 20% was used for testing it). The input features of the model were the spectrometer readings for BSS moderator spheres and the categorical feature "spectrum type" describing the facility and conditions under which the spectrum was obtained. The output parameters of the model were the spectrum description in the form of a histogram for 60 energy values, as well as the dose rate calculated from the spectrum for the corresponding conversion factor. Since the dataset of real spectra is small, database of $10^4$ synthetic data generated using the Frascati Unfolding Interactive Tool method was developed. Second model for this synthetic dataset was trainted and compared with the first one. The effect of the error in the initial data on the spectrum and the dose rate obtained from it was estimated by the Monte Carlo method using random samples. The test dataset showed that the unfolded spectra are close in nature to the original ones and have a high correlation with them. The paper proposes a method for selecting the optimal number of moderator spheres based on the explainable artificial intelligence method "Shapley additive explanation" (SHAP). The SHAP method was used to demonstrate the degree of influence of measurements with moderator spheres of different diameters on the spectrum prediction. It was shown that resulting spectrum is most influenced by measurements with moderator sphere of 10". Optimization of the choice of spheres leads to a decrease in the personnel doses during measurements. The model was trained and calculations were performed on the JINR Multifunctional Information and Computing Complex.

## 1 Introduction

High-energy neutron fields are a significant concern in nuclear power facilities, particle accelerators, and space environments due to their potential health risks to personnel. Accurate estimation of neutron energy spectra and effective dose rates is crucial for radiation protection (Aleinikov et al., 1994). However, direct measurement of the neutron spectrum, especially with energies greater than 20 MeV, is a complex experimental task, which led to the development of indirect spectrometry methods, among which the Bonner multi-sphere spectrometer (BSS) is most used (Bramblett et al.).

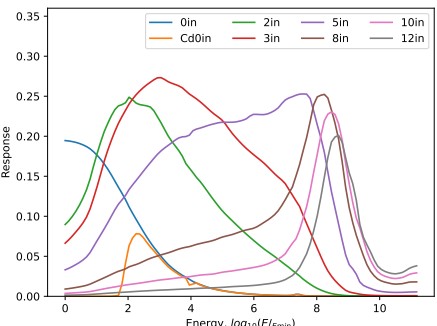

Figure 1: Response functions of Bonner multi-sphere spectrometer (Martinkovic & Timoshenko, 2005) in linear interpolation, $E_{\min} = 10^{-8}$.

But the process of unfolding neutron spectra from BSS measurements involves solving the system of ill-posed Fredholm integral equations of the first kind (Chizhov et al., 2024).

The Bonner spectrometer is based on the use of a set of polyethylene spherical moderators of different diameters. Neutrons interact with the moderator material and lose energy. Then, thermal neutrons are captured by a counter in the center of the sphere. Response of the BSS for each sphere has a maximum at a certain neutron energy value depending on moderator size. As a result the combination of different moderating spheres has a sensitivity to neutrons over a broad energy range.

## 1.1 FORMULATION OF THE PROBLEM

Neutron spectrum $\varphi(E)$ can be obtained from Bonner spectrometer measurements using an unfolding algorithm that solves a problem described by the system of $M$ Fredholm integral equations of the first kind (Chizhov et al., a;b):

$$\int_{E_{\min}}^{E_{\max}} K_j(E)\,\varphi(E)\,dE = Q_j, \quad j = 1, \dots, M, \tag{1}$$

where $Q_j$ is the Bonner spectrometer reading for the $j$-th sphere, $K_j(E)$ is the kernel of the $j$-th equation, which is a response function (RF) of the detector to neutrons of various energies, $M$ is the number of spheres used in measurements. The integration limits $E_{\min}$ and $E_{\max}$ are specified by neutron energies ($E$) and the set of detectors used. The values of $K_j(E)$ are calculated in modeling software or obtained in experiments (Martinkovic & Timoshenko, 2005). Continuous values of $K_j(E)$ are obtained by interpolation, Fig. 1.

The constraints for the problem are in the physical sense of used quantities. $Q_j \geq 0$, $\varphi(E)$ and $K_j(E)$ are smooth non-negative functions. Measurements always have errors, in this work we will assume that the detector readings contain an error within $\zeta_Q = 5\%$, and the calculation of the matrix of response functions is sufficiently accurate and the error for it can be neglected.

Fredholm integral equation of the 1st kind (eq.1) usually is solved numerically, reducing it to a system of linear algebraic equations (SLAE).We discretize eq. (1) on the grid along the energy axis ($N$ steps) and the moderator sphere diameters ($M$ steps). Matrix representation eq. (1) of after discretization:

$$\mathbf{A}\varphi = \mathbf{q} \tag{2}$$

where $\mathbf{A} \in \mathbb{R}^{M \times N}$ – kernel matrix, $\varphi \in \mathbb{R}^N$ – spectre, $\mathbf{q} \in \mathbb{R}^M$ – measurements.

However, the high correlation between the components of the kernel matrix and measurement errors leads to the instability of the solution, therefore regularization methods are used, fig. 1.1. But regularization methods require the selection of a regularization parameter, and if the choice is incorrect, the resulting spectrum may be too smooth or contain non-physically caused noise (Chizhov et al., c). Methods for selecting the optimal regularizing parameter and solution with weights for each moderator sphere in the regularizing functional are described in (Chizhov & Chizhov, a;b).

Other methods for spectrum unfolding include iterative algorithms, e.g., MAXED, GRAVEL, (Barros & et al, 2014), machine learning (ML) and neural networks (Vega-Carrillo et al., 2006). Machine

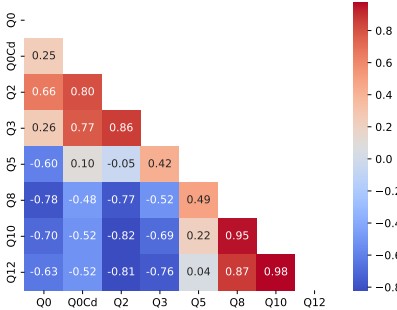

Figure 2: Correlation matrix for BSS readings.

supervised learning, has been successfully applied to the problem of neutron spectrum unfolding. These methods leverage large datasets of simulated(Bouhadida et al.), augmented (McGreivy et al.) and experimental neutron spectra with corresponding BSS responses to train models that can predict $\varphi(E)$ from $Q_i$. But not all described in literature datasets include high-energy neutron fields, which limits the applicability of those models. The results of the algorithms often are evaluated by the minimum average error or by the minimum discrepancy between the measured and reconstructed detector readings, but are not interpreted by the methods of explainable artificial intelligence (XAI). Fine-tuning of hyperparameters is sometimes also not described.

In this study, we propose a machine learning-based approach using Random Forest (RF) Regression to unfold the neutron spectra. In the context of neutron spectrum unfolding, RF offers an alternative to iterative algorithms or matrix inversion. Its ability to handle non-linear relationships and provide feature importance makes it particularly suitable for spectrum unfolding. RF was used to find $N$ values of the spectrum for $M$ input features (measurements), and to find the effective dose rate from $M$ input features.

RF is an ensemble learning method that combines multiple decision trees to improve predictive accuracy and reduce overfitting. Despite its popularity, the "black-box" nature of RF models poses significant challenges to the interpretability of the method for dosimetry purposes. XAI techniques aim to bridge this gap by providing insights into how RF models make predictions. Among the most widely used XAI methods are SHAP (SHapley Additive exPlanations) and LIME (Local Interpretable Model-agnostic Explanations) (Garreau & Luxburg, 2020). In this study SHAP was used for the evaluation of moderator spheres selection for reducing personnel exposure during measurements.

SHAP is a game-theoretic approach based on Shapley values, which originate from cooperative game theory. It assigns each feature an importance value for a specific prediction by considering all possible feature combinations. For RF models, SHAP provides consistent and locally accurate explanations, ensuring that the sum of the feature contributions equals the difference between the model's prediction and the baseline expectation (Lundberg & Lee, 2017). SHAP values are particularly useful for understanding global model behavior, as they can be aggregated across multiple instances to identify overall feature importance. However, SHAP can be computationally expensive, especially for large datasets and complex models (Molnar et al.).

## 2 METHODOLOGY

### 2.1 DATA COLLECTION AND PREPROCESSING

A database of 251 neutron spectra from various power facilities was compiled from the IAEA compendium (Com, 2001). Each spectrum was associated with readings from eight moderator spheres and a categorical feature describing the facility and measurement conditions. The data were split into training (80%) and testing (20%) sets. The database includes spectra for various power plants, which were obtained by various methods of spectrum reconstruction, as well as measured and calculated using Monte Carlo simulation software. All spectra in database were normalized to unity.

To improve algorithm's performance, a synthetic sample was also created and tested on data from the compendium. $10^4$ spectra were generated according to the FRUIT method (Bedogni et al.). In the FRUIT method, the neutron spectrum is modelled as a superposition of four component spectra described by the fission, evaporation, Gaussian, and high-energy neutron models, expr. (3).

$$\varphi(E) = P_{th}\varphi_{th}(E) + P_e\varphi_e(E) + P_f\varphi_f(E) + P_{hi}\varphi_{hi}(E), \tag{3}$$

where $\varphi_{th}(E) = (\frac{E}{T_0^2})e^{-E/T_0}$ is the Maxwell thermal component, $\varphi_e(E) = [1 - e^{-(E/E_d)^2}]E^{b-1}e^{-E/\beta'}$ is epithermal, $\varphi_f(E)$ is fast and $\varphi_{hi}(E) = (\frac{E}{T_{hi}^2})E^{-E/T_{hi}}$ – high-energy component, $P_{th}, P_e, P_f, P_{hi}$ – fraction of each component in the full spectrum ($P_{th} + P_e + P_f + P_{hi} = 1$), $T_0 = 2.53 \cdot 10^{-8}$ MeV, $E_d = 7.07 \cdot 10^{-8}$ MeV. Expression 3 allows to obtain a set of spectra simulating the energy distribution of neutron flux density at most power plants, including fission sources, radionuclide sources, medical cyclotrons and hadron accelerators. Fast neutrons are described by eq. (4) for the fission model, by eq. (5) – evaporation and by eq. (6) for Gaussian distribution.

$$\varphi_{ff}(E) = E^\alpha e^{-E/\beta} \tag{4}$$

$$\varphi_{fe}(E) = (\frac{E}{T_{ev}^2})e^{-E/T_{ev}} \tag{5}$$

$$\varphi_{fg}(E) = e^{\frac{-(E-E_m)^2}{2(\sigma E_m)^2}} \tag{6}$$

Each of these component spectra contains adjustable parameters $0 < \alpha < 1, 1 < \beta < 2, 0 < \beta' < 1, -0.5 < b < 0.5, b, \sigma, T_{ev}, T_{hi}, E_m$, which were drawn randomly during spectra generation.

Each of the 251 spectra of IAEA Compendium and $10^4$ generated spectra are represented by points on the scattering diagram (Fig. 2.1), where the ordinate axis shows the Shannon information entropy of the spectrum (Bragin et al.), and the abscissa axis shows the logarithm of the average neutron energy in the spectrum. Such a presentation of data allows us to characterize the spectra and determine the features for each type of facility.

BSS may have different sensitivity functions, since the density of polyethylene and the counter itself may differ. Therefore, it is necessary to prepare a training dataset specifically for the BSS with its response functions. According to eq. (1) for the Bonner spectrometer with the LiI(Eu) detector, the effective readings were calculated for the set of moderator spheres, with a radius of 2, 3, 5, 8, 10, 12 inch, without moderator with cadmium foil (Cd0") and without a moderator (0") using the spectra and sensitivity functions of the detector (Martinkovic & Timoshenko, 2005).

## 2.2 Random Forest Regression

The RF algorithm was used to unfold the neutron spectra. The input features were the spectrometer readings and the categorical feature, while the output parameters were the neutron energy spectrum (represented as a histogram for 60 energy values with equal grid step in logarithmic scale).

## 2.3 Dose rate assessment

In accordance with radiation safety standards (NRB, 2009), There are limits on personnel exposure to neutrons that must not be exceeded. Therefore, the effective dose rate for the isotropic (ISO) irradiation type was chosen as the criterion for comparing the unfolded spectra, eq. (7).

$$\dot{H} = \times \int_{E_{min}}^{E_{max}} \varphi(E) \cdot h(E)dE, \tag{7}$$

where $\dot{H}$ is the effective dose rate, $h(u)$ is the corresponding dose conversion coefficient (Petoussi-Henss & et al, 2010), for monoenergetic particles in a certain irradiation geometry (ISO) (Com, 2001). For each unfolded spectrum the $\dot{H}$ was calculated and the $\dot{H}$ was used as the output to train and test the model. Hyperparameter optimization was performed to improve model accuracy using `optuna` (Akiba et al., 2019).

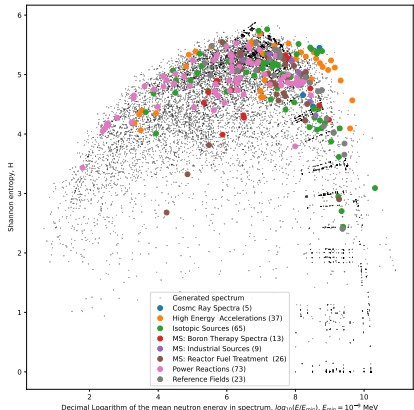

Figure 3: Scatter plot, the ordinate axis shows Shannon's information entropy, the abscissa axis shows the logarithm of neutron mean energy. The color (symbol) indicates the type of facility.

## 2.4 SHAP METHOD

The SHAP method was used to evaluate the influence of each moderator sphere on the predicted dose rate. The machine learning algorithm was used for spectrum unfolding, then the dose was estimated from the spectrum, and the contribution of each measurement to the dose was estimated using the SHAP method.

## 2.5 ERROR ESTIMATION

The Monte Carlo method was used to estimate the impact of initial data errors on the unfolded spectra and dose rates. A total of $10^4$ random samples of BSS measurements were generated to assess the robustness of the model. The detector readings were presented as an array of random numbers distributed uniformly within $\zeta_Q = 5\%$ of the true value. The dose rate obtained for all samples constitute the uncertainty region of the unfolded $\dot{H}$.

## 3 RESULTS

### 3.1 SPECTRUM UNFOLDING WITH RANDOM FOREST

The algorithm was implemented in Python using the `Scikit-learn` library. The calculations were performed on the Multifunctional Information and Computing Complex of the JINR Laboratory of Information Technologies (HybriLIT). The Random Forest algorithm was optimized over 1000 iterations using the `optuna` library, with the mean absolute error (MAE) as the optimization metric (objective value).

From input features BSS readings for moderator spheres of 2" and 12" were excluded because of high correlation with spheres of 3" and 10" respectively.

Four hyperparameters were tuned:

- the number of trees in the forest ($n\_estimators = 169$),
- the maximum tree depth ($max\_depth = 39$),
- the minimum number of samples required to split an internal node ($min\_samples\_split = 2$),
- the minimum number of samples required to be at a leaf node ($min\_samples\_leaf = 1$).

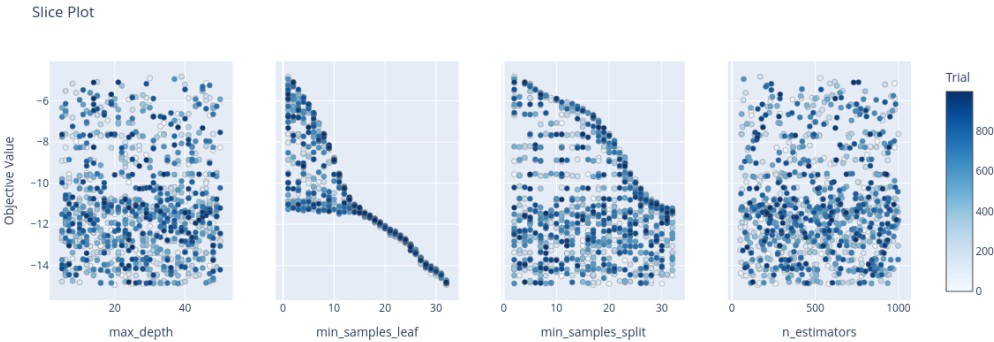

Figure 4: Optimization of the Random Forest Algorithm, $10^3$ trials.

Optimization showed that the parameter $min\_samples\_leaf$ has the main importance $94\%$ for the algorithm's performance, followed by $min\_samples\_split$ with $6\%$. The optimization results are presented in Fig. 3.1.

The model has the $R^2 score = 0.57$, mean squared error $MSE = 1.3 \cdot 10^{-3}$ and $MAE = 1.9 \cdot 10^{-2}$ for the test dataset, and $R^2 score = 0.95$, $MSE = 1.84 \cdot 10^{-4}$ and $MAE = 6.5 \cdot 10^{-3}$ for the training data. Unfolded spectres are presented in fig. 3.1. Despite the differences unfolded spectres have strong correlation with actual data $R_c = 0.84$.

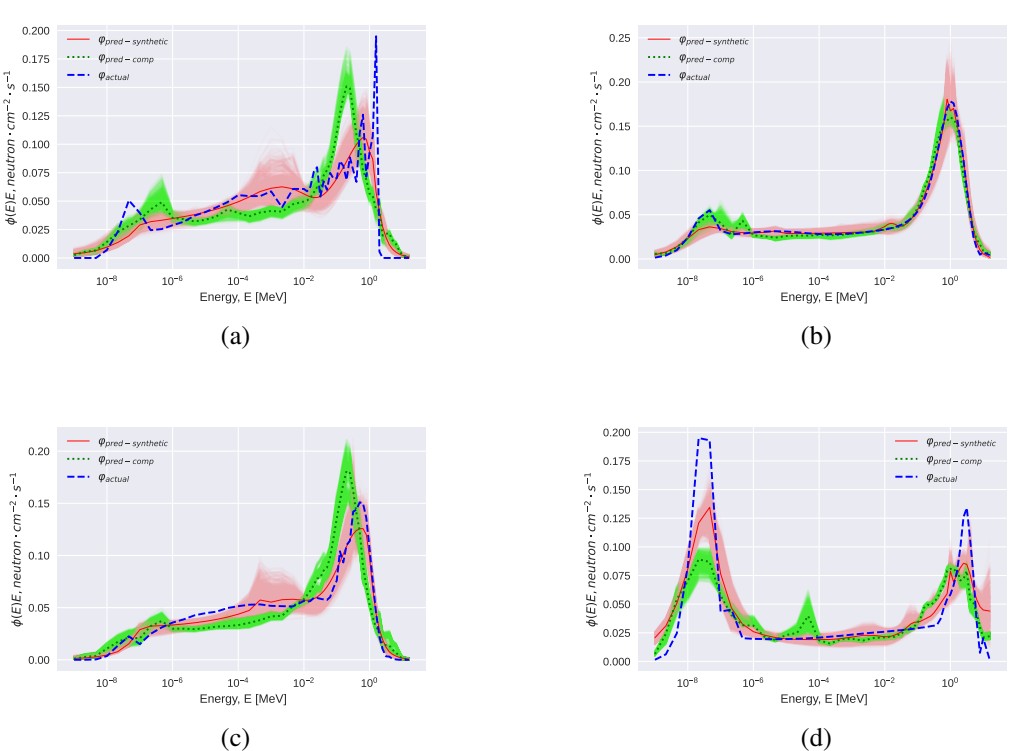

Figure 5: Comparison of unfolded and actual spectres from the test dataset. $\varphi_{pred-synthetic}$ – unfolded spectra for RF trained on synthetic dataset (transparent area shows the uncertainty), $\varphi_{pred-comp}$ – unfolded spectra for RF trained on IAEA compendium dataset, $\varphi_{actual}$ – actual spectre from IAEA compendium.

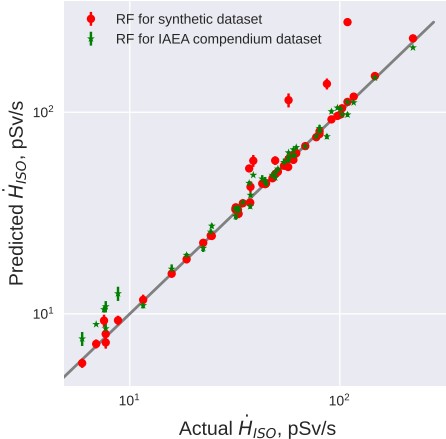

Figure 6: Comparison of predicted and actual $\dot{H}$ for the test data.

### 3.2 SPECTRUM UNFOLDING WITH RANDOM FOREST ON SYNTHETIC DATA

Second model was trained on $10^4$ synthetic spectres, obtained by FRUIT method, eq. (3). The same hyperparameters were chosen for the RF model, but as features only the measurements are taken without the categorical parameters. This model has better $R^2 score = 0.67$ for the test dataset than previous model, $MSE = 1.6 \cdot 10^{-3}$ and $MAE = 1.4 \cdot 10^{-2}$. For the training dataset metrics are: $R^2 score = 0.95$, $MSE = 2.8 \cdot 10^{-4}$ and $MAE = 5.4 \cdot 10^{-3}$. For each spectra the uncertainty for the 5% noise was calculated. Unfolded spectres are shown in figure 3.1. The uncertainty region is narrow, the solution is sufficiently robust to measurement errors. The correlation with real spectra for the spectra obtained using this model is $R_c = 0.79$.

### 3.3 EFFECTIVE DOSE RATE ASSESSMENT

Effective ISO dose rate ($\dot{H}$) uncertainty was calculated for 5% noise in BSS measurements. $\dot{H}$ was assessed for $10^4$ random samples of BSS measurements with RF algorithm. As a result a mean dose rate was calculated and the standard deviation was chosen as the dose rate estimation error. Results are shown in figure 3.3. For the test data of 51 spectres from IAEA compendium both models showed good results. For most spectra, the assessed effective dose rate coincided within a few percent with the value for the original spectrum. Detailed comparison is the figure 3.3.

### 3.4 SHAP ANALYSIS

The SHAP method implemented as a Python library was applied to the random RF to interpret its work for the unfolded $\dot{H}$. RF model for BSS measurements as features and $\dot{H}$ as the target was developed. For each measurements the local interpretation is original. According to the Petoussi-Henss & et al (2010) high energy neutrons give the most contribution to the dose. The local interpretation confirms this, showing that measurements with the 10" diameter moderator sphere measurements make the major contribution to the dose rate prediction, fig. 3.4. Due to the correlation of response functions, if sphere 10" is replaced by sphere 12" then the assessed $\dot{H}$ will be almost the same.

By summing up the coefficients for the entire sample, we obtain a global interpretation of the influence of all the model features on the final result. The global interpretation confirms that for all test subdataset the greatest weight in the $\dot{H}$ was contributed by measurements with a 10" moderator sphere. Other spheres are responsible for sensitivity to neutrons of lower energies that contribute less to the dose rate. The type of facility also has a weak affect the final result. This could be seen in the scattering diagram where there were no separated clusters to divide the spectra by facility type, 2.1. Extended view of how the top features in a dataset impact the model's output is shown in the beeswarn plot, fig. 3.4.

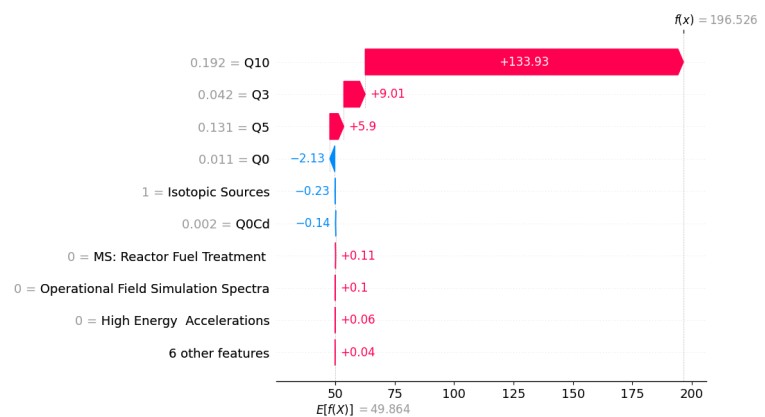

Figure 7: Local interpretation with SHAP for one spectre from test dataset.

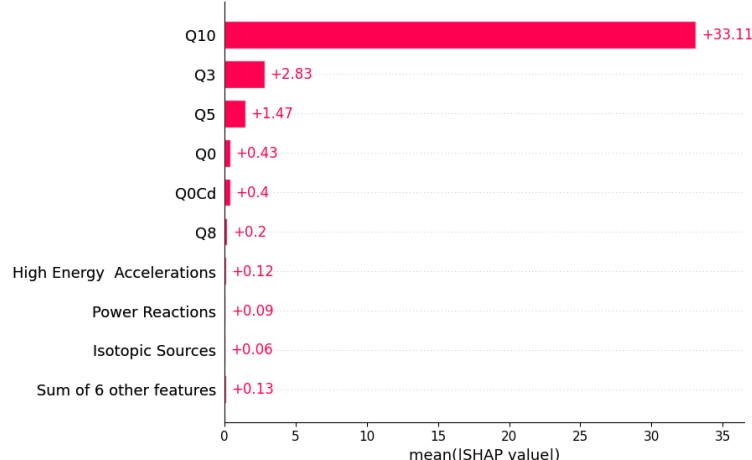

Figure 8: Global interpretation with SHAP.

The SHAP analysis revealed that measurements with 10" moderator spheres had the most significant impact on spectrum prediction. This finding allowed for the optimization of the measurement setup, reducing the number of required spheres and minimizing personnel exposure.

## 4 DISCUSSION

The proposed method demonstrates the potential of machine learning for neutron spectrum unfolding and dose rate estimation. The use of Random Forest Regression provides a robust and computationally efficient alternative to traditional methods. The SHAP analysis further enhances the practicality of the approach by optimizing the measurement setup. However, the method's performance may be limited by the quality and diversity of the training data.

The advantages of RF for spectrum unfolding are as follows: due to the ensemble nature of RF, it is less prone to overfitting compared to single decision trees or other machine learning models, especially when dealing with small datasets. RF can capture complex, non-linear relationships between the input features (Bonner sphere readings) and the output (neutron spectrum or dose rate), making it suitable for solving ill-posed problems like spectrum unfolding. RF provides intrinsic feature importance scores, which can be used to identify the most influential moderator spheres or measurement conditions. RF does not require input features to be normalized or scaled, simplifying the preprocessing pipeline. RF can handle missing values in the input data, which is beneficial when dealing with incomplete experimental data.

The disadvantages of RF for spectrum unfolding with small Datasets is known well. RF performs poorly when predicting outside the range of the training data. For spectrum unfolding, this can be problematic if the test data contains energy ranges or conditions not represented in the training set.

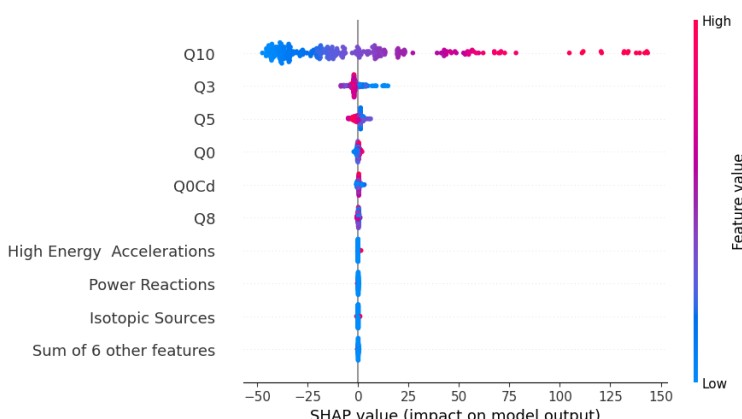

Figure 9: Beeswarn plot for the global interpretation with SHAP.

Although RF is robust to overfitting, its performance can still degrade with very small datasets. The model may struggle to learn the underlying patterns if the training data is insufficiently diverse or representative. The performance of RF depends on hyperparameters such as the number of trees, maximum depth, and the number of features considered at each split. Optimizing these hyperparameters can be challenging with limited data.

The proposed method could be used for unfolding neutron spectra in stationary fields. However, the accuracy of the reconstructed spectra heavily depends on the quality of the response functions. If the spectra in the training dataset were obtained by solving the inverse problem using regularization methods, the reliability of the unfolded spectra may be compromised. Additionally, the method's performance is constrained by the specific set of moderator spheres used in the BSS. For instance, due to the nature of the response functions, measurements from moderator spheres with close diameters are highly correlated, which can negatively impact the method's results.

To improve the algorithm's performance, it is essential to expand the training dataset. This can be achieved through data augmentation techniques as well as by incorporating real spectra unfolded using alternative methods. Furthermore, spectra calculated using Monte Carlo simulation software can be added to the dataset to enhance its diversity and representativeness.

When comparing the results of spectrum unfolding methods, it is possible to use alternative metrics of the discrepancy between the true and unfolded distributions: the Kolmogorov-Smirnov statistic (Bogomolov et al., 2024) or maximum mean discrepancy (Gretton et al., 2012; Nguyen et al., 2020).

As a future work automated ML could be used to find the best set of ML algorithms, like *LightAutoML* (Vakhrushev et al., 2021).

## 5 CONCLUSION

This study presents a machine learning-based approach for neutron spectrum unfolding and effective dose rate estimation using Bonner spectrometer measurements. Comparison of two Random Forest Regression models trained on a small real spectres dataset and on a large synthetic dataset is presented. Both models showed strong correlation between actual and unfolded spectra and in dose rate assessment. The SHAP method enabled the optimization of BSS moderator sphere selection, it is showed that measurements with the 10" moderator sphere make are the most important to the dose assessment. The proposed method could be used for improving radiation protection in high-energy neutron fields.

## ACKNOWLEDGMENTS

The authors acknowledge the support of the JINR Multifunctional Information and Computing Complex for providing computational resources. The research was carried out within the state assignment

of Ministry of Science and Higher Education of the Russian Federation (theme No. 124092700007-4).

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
