# OpenReview forum: "Random forest regression and Shapley additive explanation for effective dose rate estimation in high-energy neutron fields based on Bonner spectrometer measurements"
_mathai.club/MathAI/2025/Conference — MathAI 2025 Oral_

### Official Review · Reviewer_5xkx · 2025-02-25
**Random forest regression and Shapley additive explanation for effective dose rate estimation in high-energy neutron fields based on Bonner spectrometer measurements**

**Rating:** 8
**Confidence:** 4

**Review:**

The paper proposes a method for assessing the neutron energy
 spectrum and effective dose of personnel based on the readings
of a Bonner spectrometer for high-energy neutron fields.
The spectra were unfolded using the algorithm ”random forest”
with optimization of the model hyperparameters. The paper also
 proposes a method for selecting the optimal number of moderator
 spheres based on the explainable artificial intelligence
 method ”Shapley additive explanation” (SHAP).

Overall, the work is certainly of interest as an attempt to
introduce a machine learning-based approach for neutron spectrum
unfolding and dose rate estimation using Bonner spectrometer measurements.

In what follows, some remarks and comments in order to improve the work

1. The paper lacks a comprehensive reference study of the research
topic despite its importance and the previous studies that were
 presented are very few. Only two references were mentioned,
one in 1994 and the other in 2024, and the gap between them is
20 years.

2. The referring to fig. 1.1 is not understandable. In general
the reference of the figures in the text when referring to
any figure is not consistent with the number of the figure in
its caption

There are many typos in the paper and in the following some of them
1. line 018. In our paper  ----->  In our paper,
2. line 018. using supervised machine  -----> using the supervised machine
3. line 034. that resulting  -----> that the resulting
4. line 035. with moderator sphere -----> with a moderator sphere
5. line 051. and loose energy -----> and lose energy
6. Last line on page 1.  has sensitivity -----> has a sensitivity
7. line 091. However, high correlation -----> However, the high correlation
8. line 092.  lead to instability  -----> leads to the instability
9. line 099.  have been successfully -----> has been successfully
10. line 129. for dosimetry puposes. -----> for dosimetry purposes.
11. line 128. for interpretability -----> to the interpretability
12. line 132. used for evaluation  -----> used for the evaluation
13. line 199. method used for spectrum  -----> method was used for spectrum
14. line 364. Advantages of RF-----> The advantages of RF
15. line 373. Disadvantages of RF -----> The disadvantages of RF
16  line 414. it is showed that measurements -----> it showed that measurements

---

### Official Review · Reviewer_V6r7 · 2025-02-25
**Physics-related paper with less clear ML-related part**

**Rating:** 6
**Confidence:** 3

**Review:**

The paper proposes an approach to predict the characteristics of high energy neuron fields using random prediction regression.

The domain problem is well stated and the description of the problem statement and the experimental setup is clear.

However, the ML part is less clear. Firstly there is no explanation of why the RF model was used, nor is there any comparison with other models (e.g. different gradient boosting models) or AutoML tools.

Some figures (Fig.5-fig.7) are not mentioned at all in the text, while some references as "Fig.3.2" lead to non-existing figures. This may be a technical error in LATEX.

The prediction results presented in Figure 5 look like the prediction error is very high. This problem should be discussed in the text (or the meaning of the figure should be explained). It is not clear to me why the accuracy is high here.

So the paper should be improved to make the machine learning related part of the paper clearer. It would also be useful to provide source code for experiments to increase the reproducibility. At the double-bling review stage, https://anonymous.4open.science/ can be used.

---

### Official Review · Reviewer_uGqi · 2025-02-27
**Essence**

**Rating:** 6
**Confidence:** 3

**Review:**

Essence:
1 The model's hyperparameters are not described (lines 252-257);

2 There is no justification for the choice of metrics used to evaluate prediction quality. In
line 104, the author notes that the minimum discretion metric is referenced in various
papers addressing the problem. Why has the author not calculated this metric?

3 There is no rationale provided for the selection of features used to train the
RandomForest model;

4 The title of the paper indicates that the work involves the use of RandomForest for
forecasting and SHAP analysis to elucidate the forecast.

However, the main body of results and discussions primarily focuses on the description
of the chosen machine learning model. Furthermore, in lines 097-105, the authors state
that various ML models and neural networks have already been developed to address the
problem at hand. Consequently, the advantages of employing RandomForest for
forecasting remain unclear, as there is no comparison with the metrics of existing models
designed to tackle the issue.

If the primary novelty of the authors' research lies in conducting a SHAP analysis using a
trained model (as described in lines 097-105), a more detailed account of the SHAP
analysis conducted in the results and discussion is warranted.

5 In line 365, the author asserts that RandomForest is less prone to overfitting; however,
this remains a possibility with ensemble methods. The article lacks justification for the
absence of retraining in addressing this issue.

6 The author does not explain the significant increase in the MSE metric on the test dataset.

7 In lines 371-372, the author states that the RandomForest model effectively manages
missing values. However, the skip processing methods implemented in RandomForest
may not be appropriate for addressing this issue. A description of why these skip
processing methods are suitable for solving the problem is necessary. If the author
employed their own methods for handling omissions, this should also be detailed.

Decoration:

1 The reference numbers for the figures do not correspond to their actual numbers.

---

### Official Review · Reviewer_wmqW · 2025-02-27
**Interesting application of explainable AI, but methodology needs improvement**

**Rating:** 6
**Confidence:** 4

**Review:**

Neutron spectra unfolding represents a significant area of current research due to the critical relationship between neutron energy and radiation dosimetry. This paper presents a novel application of Shapley Additive Explanations (SHAP) to analyze feature importance in neutron spectrometry, particularly examining various moderator spheres' contributions to dose rate prediction. To my knowledge, this represents the first implementation of SHAP values in this specific domain.

Despite this innovative approach, several methodological concerns from a machine learning perspective warrant attention

*Limitations in Dataset Size and Generalizability*

The study utilizes only 251 spectra, which raises significant concerns about statistical power and model robustness. Machine learning models, particularly ensemble methods like RF, perform better with large and diverse datasets. The small dataset size may lead to overfitting and reduced generalizability. The reported performance metrics (R² = 0.997 for training versus R² = 0.989 for testing) suggest potential overfitting, as this considerable gap indicates the model may be memorizing training examples rather than learning generalizable patterns. Authors should Implement k-fold cross-validation to better assess generalizability and mitigate overfitting risks.

*Lack of Alternative ML Comparisons*

The paper does not benchmark RF performance against other ML models such as Neural Networks, Gradient Boosting (e.g., XGBoost), or Bayesian methods. A comparative study would help establish RF’s relative efficacy.

*Feature Engineering and Selection Methodology*

While the selection of moderator sphere readings and spectrum type as input features appears reasonable, the justification for excluding certain spheres requires more rigorous explanation. The research could be strengthened by incorporating dimensionality reduction techniques such as Principal Component Analysis (PCA) to complement the SHAP analysis and provide more comprehensive feature importance insights.

*Error Propagation and Uncertainty Quantification*

The study reports a mean relative error of 76% for data with 5% uncertainty, which is significantly high. This suggests that small errors in measurement inputs can drastically affect dose rate predictions, reducing reliability in real-world applications. The notable discrepancies visible in Figure 6 further amplify this concern.

*Dependence on Regularization Methods for Training Data*

The authors acknowledge deriving some training data through regularization-based inverse problem-solving methods. This introduces a potential methodological weakness, as the machine learning model may inadvertently propagate approximation errors present in the training dataset rather than correct for them.

*Conclusion*

The paper presents a valuable application of machine learning to neutron spectrometry, but it faces challenges related to dataset size, overfitting, interpretability, and uncertainty. Addressing these concerns through expanded datasets, model benchmarking, and refined feature selection will enhance the robustness and applicability of the proposed method.

---

### Decision · Program_Chairs · 2025-03-08

**Decision:**

Accept (Oral)

**Comment:**

Your article has been accepted and you can make a presentation on the article. All articles will be sorted by rating and within the available conference places one author from each article will be invited. If there are not enough places, then you will either have the opportunity to present remotely or come at your own expense!